# Sequential cocatalyst decoration on BaTaO$_2$N towards highly-active Z-scheme water splitting

Zheng Wang [1,2,9], Ying Luo[3,9], Takashi Hisatomi [1], Junie Jhon M. Vequizo [1], Sayaka Suzuki [4], Shanshan Chen [1], Mamiko Nakabayashi [5], Lihua Lin[1], Zhenhua Pan[1], Nobuko Kariya[6], Akira Yamakata [7], Naoya Shibata [5], Tsuyoshi Takata[1], Katsuya Teshima [1,4✉] & Kazunari Domen [1,8✉]

Oxynitride photocatalysts hold promise for renewable solar hydrogen production via water splitting owing to their intense visible light absorption. Cocatalyst loading is essential for activation of such oxynitride photocatalysts. However, cocatalyst nanoparticles form aggregates and exhibit weak interaction with photocatalysts, which prevents eliciting their intrinsic photocatalytic performance. Here, we demonstrate efficient utilization of photoexcited electrons in a single-crystalline particulate BaTaO$_2$N photocatalyst prepared with the assistance of RbCl flux for H$_2$ evolution reactions via sequential decoration of Pt cocatalyst by impregnation-reduction followed by site-selective photodeposition. The Pt-loaded BaTaO$_2$N photocatalyst evolves H$_2$ over 100 times more efficiently than before, with an apparent quantum yield of 6.8% at the wavelength of 420 nm, from a methanol aqueous solution, and a solar-to-hydrogen energy conversion efficiency of 0.24% in Z-scheme water splitting. Enabling uniform dispersion and intimate contact of cocatalyst nanoparticles on single-crystalline narrow-bandgap particulate photocatalysts is a key to efficient solar-to-chemical energy conversion.

[1] Research Initiative for Supra-Materials, Interdisciplinary Cluster for Cutting Edge Research, Shinshu University, Nagano-shi, Nagano, Japan. [2] Research Center for Eco-Environmental Sciences, Chinese Academy of Sciences, Beijing, China. [3] Department of Science and Technology, Graduate School of Medicine, Science and Technology, Shinshu University, Nagano, Japan. [4] Department of Materials Chemistry, Faculty of Engineering, Shinshu University, Nagano, Japan. [5] Institute of Engineering Innovation, The University of Tokyo, Tokyo, Japan. [6] Science & Innovation Center, Mitsubishi Chemical Corporation, Yokohama-shi, Kanagawa, Japan. [7] Graduate School of Engineering, Toyota Technological Institute, Nagoya, Japan. [8] Office of University Professors, The University of Tokyo, Tokyo, Japan. [9] These authors contributed equally: Zheng Wang, Ying Luo. ✉email: teshima@shinshu-u.ac.jp; domen@shinshu-u.ac.jp

Water splitting using particulate photocatalysts is regarded as a technologically simple and cost-competitive approach toward sustainable solar hydrogen production owing to its potential for large-scale applications[1–5]. However, the solar-to-hydrogen energy conversion efficiency (STH) of particulate photocatalysts in water-splitting processes is still behind those of photovoltaic or photoelectrochemical devices[6,7]. Nevertheless, recent improvements in quantum efficiency, the optical wavelength range usable for the reaction, and the STH of particulate photocatalysts and reactors[8–13] are encouraging for the development of efficient particulate photocatalysts with narrow bandgaps for water splitting. In particular, the Z-scheme water-splitting system has an advantage for harvesting visible light in a wide wavelength range, because it utilizes two-step photoexcitation of an $H_2$-evolving photocatalyst (HEP) and an $O_2$-evolving photocatalyst (OEP), and therefore allows to work with versatile narrow-bandgap particulate photocatalysts[10,14]. However, existing Z-scheme systems exhibit low water-splitting efficiency, even though 600-nm-class (oxy)nitrides[15–17], (oxy)chalcogenides[18–20] and dye-sensitized photocatalysts[21] have been applied as HEPs (see Supplementary Table 1). To take advantage of the Z-scheme system, it is essential to identify the factors needed to activate narrow-bandgap photocatalyst materials.

Perovskite-type $BaTaO_2N$ with a bandgap of around 1.8 eV is a photocatalyst material that has been intensively studied for Z-scheme water splitting[15,16,22,23], as well as $H_2$ or $O_2$ evolution half-reactions using sacrificial reagents[15,24–27] (see Supplementary Table 2). The $BaTaO_2N$ photocatalyst intrinsically exhibits a weak driving force for surface redox reactions[14], and loading of cocatalysts such as nanoparticulate Pt is essential to promote the extraction of photogenerated charge carriers from $BaTaO_2N$ for efficient $H_2$ evolution[22,23,28–33]. However, Pt loaded by conventional impregnation or photodeposition methods tends to form aggregates on $BaTaO_2N$ and to have weak contact with $BaTaO_2N$ particles, resulting in the inadequate formation of active catalytic sites and inefficient electron transfer. In addition, $BaTaO_2N$ particles produced by thermal nitridation are generally polycrystalline and incorporate structural defects and midgap states that act as recombination and trapping centres for photogenerated electron–hole pairs[15,16,22,25]. Therefore, the efficiency of photocatalytic $H_2$ evolution on $BaTaO_2N$ needs to be boosted by utilizing a single-crystalline particulate photocatalyst and establishing a strategy to realize an active cocatalyst/photocatalyst reciprocal structure. Herein, we present an effective cocatalyst engineering strategy based on the stepwise deposition of Pt nanoparticles on a single-crystalline $BaTaO_2N$ particulate photocatalyst, involving an impregnation–reduction pre-treatment and a subsequent photodeposition process. In the impregnation–reduction process, uniformly dispersed Pt seeds form an intimate contact with $BaTaO_2N$. Subsequently, photodeposition allows Pt nanoparticles to uniformly grow at the numerous active Pt seeds. This approach causes the photocatalytic efficiency of the resulting Pt-loaded $BaTaO_2N$ for $H_2$ evolution from an aqueous methanol solution and Z-scheme water splitting constructed with $WO_3$ to be highly improved.

## Results and discussion

**Preparation of Pt-modified $BaTaO_2N$.** The preparation of $BaTaO_2N$, loading of Pt nanoparticles on $BaTaO_2N$, and the photocatalytic $H_2$ production and Z-scheme water splitting, are described in detail in the subsections of "Methods". Briefly, $BaTaO_2N$ was synthesized by one-pot nitridation of a $BaCO_3$ and $Ta_2O_5$ mixture with the assistance of molten RbCl or other alkali chlorides (NaCl, KCl, or CsCl) fluxes at 1223 K for 8 h under a flow of gaseous $NH_3$ (200 mL min$^{-1}$)[24,34]. $BaTaO_2N$ prepared by using RbCl, NaCl, KCl, and CsCl fluxes are denoted as $BaTaO_2N$ (RbCl), $BaTaO_2N$ (NaCl), $BaTaO_2N$ (KCl), and $BaTaO_2N$ (CsCl), respectively.

The two-step cocatalyst modification procedure was initiated by loading a small amount of the Pt precursor on $BaTaO_2N$ and a subsequent reduction treatment in an $H_2$ atmosphere at 473 K for 1 h. An additional amount of Pt cocatalyst was then deposited on the Pt-impregnated $BaTaO_2N$ by photoreduction using methanol as a sacrificial electron donor. Pt-loaded $BaTaO_2N$ samples are hereafter denoted as Pt($x$IMP + $y$PD)/$BaTaO_2N$, where $x$ and $y$ express the loaded amounts of Pt cocatalyst in weight percent by the impregnation–reduction and photodeposition processes, respectively. For comparison, a Pt cocatalyst was also loaded on $BaTaO_2N$ by an impregnation–reduction process or a photodeposition process alone, and these samples are denoted as Pt($x$IMP)/$BaTaO_2N$ and Pt($y$PD)/$BaTaO_2N$, respectively.

The materials properties of $BaTaO_2N$ (RbCl) were examined because $BaTaO_2N$ (RbCl) exhibited greater $H_2$ evolution activity than $BaTaO_2N$ synthesized by using the other alkali chloride fluxes when being decorated with Pt by photodeposition[24,34]. The X-ray diffraction pattern (Supplementary Fig. 1a) shows that typical perovskite-type $BaTaO_2N$ was obtained through RbCl flux-assisted one-pot nitridation. The UV–vis diffuse reflectance spectrum (Supplementary Fig. 1b) demonstrates a light absorption edge at 650 nm, which is characteristic of $BaTaO_2N$. The background absorption beyond 650 nm is negligible owing to the low concentration of reduced $Ta^{5+}$ species and anion vacancies[22,23]. The $BaTaO_2N$ was composed of cuboid particles smaller than 500 nm in size, as indicated in the scanning electron microscopy (SEM) image (Supplementary Fig. 1c). High-resolution transmission electron microscopy (HRTEM) images of the $BaTaO_2N$ particle are presented in Supplementary Fig. 1d and e, together with a structural model of $BaTaO_2N$ that correlates the image spots with atomic positions (Supplementary Fig. 1f). The atomic-scale periodical image contrast without any dislocations or grain boundaries indicates that $BaTaO_2N$ (RbCl) consists of single-crystalline particles.

**$H_2$-evolution activity of Pt-modified $BaTaO_2N$ (RbCl).** The $H_2$-evolution activity of the $BaTaO_2N$ (RbCl) photocatalyst was evaluated after modification with the Pt cocatalyst. Figure 1a shows the dependence of the $H_2$-evolution rate on the amount of Pt cocatalyst loaded on $BaTaO_2N$ via the different methods. The $H_2$-evolution rate was obtained during the first hour of the photocatalytic reaction in a sacrificial methanol aqueous solution (Supplementary Fig. 2). $BaTaO_2N$ loaded with a Pt cocatalyst by impregnation–reduction exhibited a more than ten times higher activity for $H_2$ evolution than those loaded with a Pt cocatalyst by photodeposition for the same loading amounts. Moreover, a remarkable enhancement of photocatalytic $H_2$ production was realized on $BaTaO_2N$ by two-step decoration of the Pt cocatalyst (Fig. 1a), in which 0.1 wt% Pt cocatalyst was loaded on $BaTaO_2N$ by the impregnation–reduction procedure and the subsequent photodeposition of additional Pt cocatalyst. The $H_2$-evolution activity increased with increasing content of photodeposited Pt cocatalyst, reaching a maximum at 0.2 wt% additional Pt loading by photodeposition (0.3 wt% Pt in total), and then sharply decreased with further Pt loading (Fig. 1a). The apparent quantum yield (AQY) for the optimized Pt-loaded $BaTaO_2N$ by sequential decoration during photocatalytic $H_2$ evolution as a function of the irradiation wavelength is plotted in Fig. 1b. The onset irradiation wavelength for $H_2$ generation agreed well with the absorption edge for this $BaTaO_2N$ photocatalyst. The AQY values were 6.8 ± 0.5% at 420 nm (±25 nm), 2.9 ± 0.4% at 500 nm (±25 nm), and 0.8 ± 0.3% at 600 nm (±25 nm), which are more than 100 times higher than those reported for $BaTaO_2N$ photocatalysts (see Supplementary Table 2). This is the most efficient photocatalytic $H_2$ evolution from a sacrificial methanol solution using a 600-nm-class photocatalyst. It should be also noted that

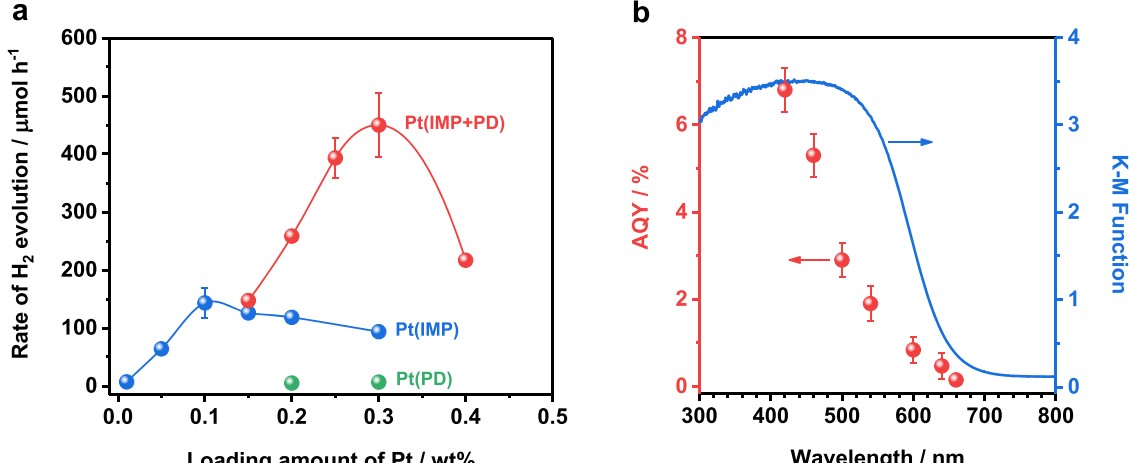

**Fig. 1 Photocatalytic H₂ evolution and apparent quantum yield of Pt-modified BaTaO₂N (RbCl). a** H₂ evolution rates as a function of the total Pt loading during visible-light-driven H₂ production reactions from an aqueous methanol solution. IMP and PD denote Pt loading by impregnation–reduction and photodeposition, respectively, and IMP + PD denotes 0.1 wt% Pt loading by impregnation–reduction and additional Pt loading by photodeposition. **b** Apparent quantum yield as a function of the incident light wavelength during visible-light-driven H₂ production over Pt(0.1%IMP + 0.2%PD)/BaTaO₂N. Conditions: Pt-modified BaTaO₂N (RbCl) photocatalyst, 0.1 g; 10 vol% aqueous methanol solution, 150 mL; light source, 300 W Xenon lamp equipped with a cut-off filter ($\lambda \geq 420$ nm) for **a** and various band-pass filters for **b**; reaction system, Pyrex top-illuminated vessel connected to the closed gas-circulation system without evacuation of gas products. Error bars indicate standard deviation for three measurements.

BaTaO₂N (RbCl) modified with Pt cocatalyst by two-step decoration was stable during the photocatalytic H₂ evolution reaction (Supplementary Fig. 3).

**Interaction of Pt cocatalyst with BaTaO₂N (RbCl).** To understand the enhancement mechanism for photocatalytic activity upon two-step cocatalyst decoration, the morphology and fine structure of Pt particles on BaTaO₂N were compared by SEM (Fig. 2a and Supplementary Fig. 4) and HRTEM (Fig. 2b). The actual loading amounts of the Pt cocatalyst on BaTaO₂N by the three deposition procedures were in accordance with the introduced Pt content in the samples (Supplementary Table 3). Pt nanoparticles of uniform size were dispersed homogeneously on each BaTaO₂N particle without obvious aggregation when 0.3 wt% Pt was loaded by the two-step decoration method. They had hemispherical shapes and intimate interfacial contact with the BaTaO₂N. Pt particles having hemispherical shapes were also deposited firmly on BaTaO₂N by the impregnation–reduction method owing to the thermal H₂-reduction treatment. However, Pt nanoparticles were aggregated on BaTaO₂N for a Pt loading of 0.3 wt%, and it was necessary to decrease the loading amount to 0.1 wt% to avoid aggregation. When the Pt cocatalyst was loaded by the photodeposition method, regardless of the cocatalyst amount (0.2 or 0.3 wt%), the Pt particles were localized at electron-accumulating sites of BaTaO₂N and formed large clusters on some active BaTaO₂N particles. Moreover, the Pt nanoparticles were spheroidal and had a small contact area with the cuboid-like BaTaO₂N, suggesting a weak interaction with the BaTaO₂N photocatalyst. Notably, X-ray photoelectron spectroscopy (XPS) spectra demonstrated that the Pt cocatalysts deposited by the three decoration methods were metallic and the surface components of BaTaO₂N were unchanged during the different cocatalyst modification procedures (Supplementary Fig. 5). Therefore, the morphology and dispersivity of Pt nanoparticles using the different decoration procedures are most likely responsible for the distinct H₂-evolution activity of BaTaO₂N.

Transient absorption spectroscopy (TAS) was used to examine how the structure of the nanoparticulate Pt cocatalyst affected the behavior of photogenerated charge carriers in BaTaO₂N (Fig. 3a).

The faster decay of the absorption intensity at 5000 cm⁻¹ (2000 nm, 0.62 eV) for the Pt-modified BaTaO₂N samples than for the bare BaTaO₂N reflects the efficient electron transfer from BaTaO₂N to the Pt cocatalyst[35,36]. Supplementary Table 4 lists the results of a quantitative estimation of the remaining electrons in the Pt-modified BaTaO₂N with respect to pristine BaTaO₂N. At 300 μs after photoexcitation, nearly 80% of photoexcited electrons in BaTaO₂N were captured by the Pt cocatalyst loaded via the two-step procedure, which was more efficient than the case for those loaded via the individual impregnation–reduction or photodeposition methods. In addition, the prolonged lifetime of photoexcited holes for Pt-loaded BaTaO₂N by two-step decoration (Supplementary Fig. 6) indicates a reduction in the electron–hole recombination rate in BaTaO₂N owing to efficient electron transfer to the Pt cocatalyst[35].

The sequential decoration method apparently avoids the problems with the individual impregnation–reduction and photodeposition methods where the Pt cocatalysts aggregate and interact weakly with the BaTaO₂N photocatalyst (Supplementary Fig. 7). The pre-loading of Pt nuclei in the initial impregnation–reduction step is regarded as a surface pre-treatment of BaTaO₂N that helps to create finely dispersed active sites to induce the uniform growth of Pt particles during the subsequent photodeposition process (Fig. 3b), because the photodeposition process still generated aggregated Pt particles on BaTaO₂N after the H₂-reduction treatment without impregnation of Pt species (Supplementary Fig. 8). This is also consistent with a control experiment in which Pd species was photodeposited as a probe. The photodeposition of Pd occurred more rapidly on Pt-impregnated BaTaO₂N than on pristine BaTaO₂N, and Pd particles were preferentially formed on the pre-introduced Pt sites (Supplementary Fig. 9). As a result, the sequential decoration method that realizes well-distributed Pt catalytic sites with modest particle sizes and intimate interaction with BaTaO₂N, can maximize electron extraction and transfer for surface proton-reduction reactions[37–41].

**Effects of the quality of BaTaO₂N photocatalysts.** We note that the properties of BaTaO₂N also play a pivotal role in achieving efficient photocatalytic H₂ evolution through two-step Pt cocatalyst

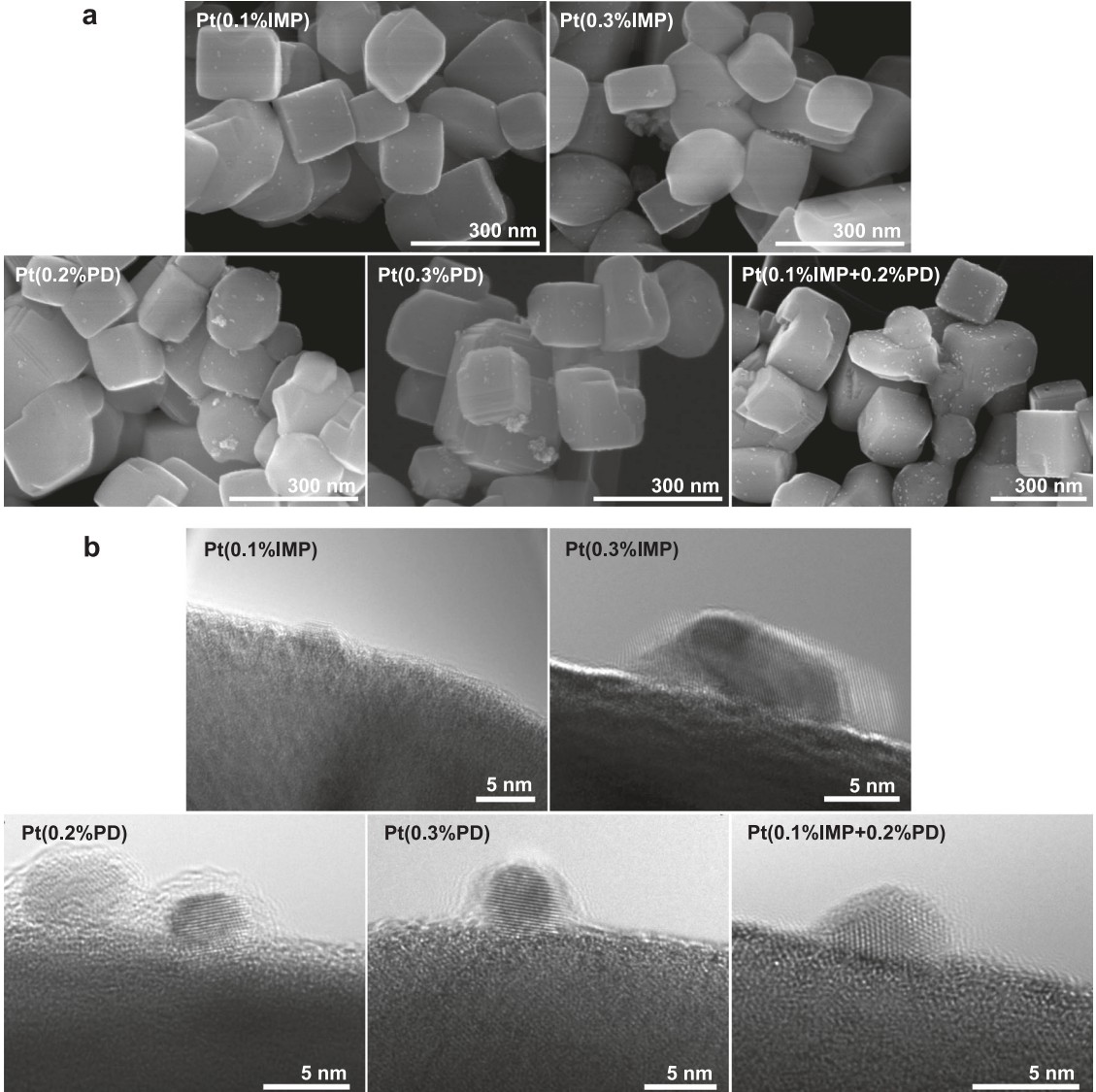

**Fig. 2 Morphology and interfacial structure of Pt nanoparticles and BaTaO$_2$N (RbCl) photocatalyst. a, b** SEM images (**a**) and HRTEM images (**b**) of Pt-modified BaTaO$_2$N produced by three different methods. IMP, PD, and IMP + PD denote Pt loading by impregnation–reduction, photodeposition, and sequential deposition, respectively.

decoration. BaTaO$_2$N (RbCl) exhibited superior photocatalytic H$_2$-evolution activity to those prepared using other alkali chloride fluxes (Fig. 4a), although these flux-assisted BaTaO$_2$N photocatalysts exhibited similar cuboid crystal shape (Fig. 2a and Supplementary Fig. 10) and the loading of the same amount of metallic Pt cocatalysts were confirmed (Supplementary Fig. 11). Figure 4b, c depicts transient absorption intensity profiles at 11000 cm$^{-1}$ (910 nm, 1.36 eV) and at 15400 cm$^{-1}$ (649 nm, 1.91 eV), reflecting the dynamics of deeply trapped electrons and photoexcited holes, respectively, acquired for these Pt-modified BaTaO$_2$N samples. The populations of deeply trapped electrons and photoexcited holes decayed more rapidly and slowly, respectively, on a microsecond timescale as the H$_2$ evolution activity became greater. From a comparison of the bare and Pt-modified BaTaO$_2$N samples (Supplementary Fig. 12a, b), BaTaO$_2$N (RbCl) benefitted most from the Pt cocatalyst loading, indicating more efficient electron migration to Pt and less electron trapping in BaTaO$_2$N. Moreover, BaTaO$_2$N (RbCl) exhibited faster decay of deeply trapped electrons than BaTaO$_2$N (NaCl) and BaTaO$_2$N (CsCl), and was similar to

that for BaTaO$_2$N (KCl) (Supplementary Fig. 12c). From the comparison of XRD patterns (Supplementary Fig. 13a), background absorption in UV–vis DRS (Supplementary Fig. 13b), and XPS analysis (Supplementary Fig. 11), RbCl flux-assisted nitridation process enabled the formation of well-crystallized BaTaO$_2$N with minimized defects (reduced Ta$^{5+}$ species and anion vacancies)[24,34]. Moreover, the amount of alkali metal ions incorporated in the BaTaO$_2$N materials during the flux-assisted nitridation became less with the increase of their ionic radii (Supplementary Table 5 and Supplementary Fig. 11). The incorporation of alkali metal ions would induce charge imbalance and defects in the BaTaO$_2$N material, thus leading to less efficient photoexcited charge carrier transfer. These results indicate that the low density of structural defects and mid-gap states in BaTaO$_2$N (RbCl) were key to suppressing charge recombination and facilitating the transfer of photoexcited electrons to the Pt cocatalyst for efficient H$_2$ evolution. In contrast, uniform dispersion of Pt nanoparticles on BaTaO$_2$N (NaCl) and BaTaO$_2$N (CsCl) photocatalysts was not realized by photodeposition even if Pt nuclei were introduced as electron-

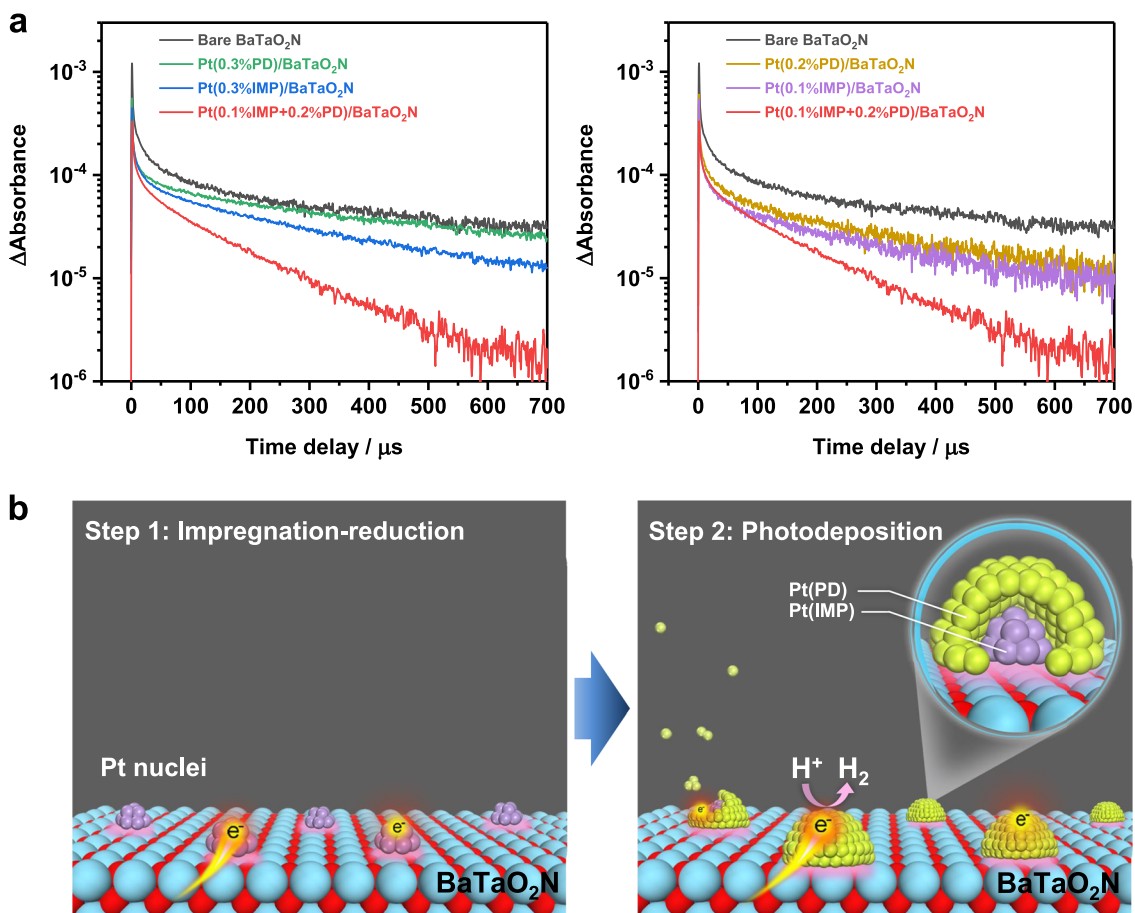

**Fig. 3 Interaction of Pt nanoparticles and BaTaO$_2$N (RbCl) photocatalyst. a** Transient absorption decays corresponding to electron dynamics in bare BaTaO$_2$N and Pt-modified BaTaO$_2$N photocatalysts probed at 5000 cm$^{-1}$ (2000 nm, 0.62 eV) at 0–700 μs. **b** Schematic of sequential Pt cocatalyst deposition on BaTaO$_2$N.

extracting sites by the first impregnation–reduction treatment (Supplementary Fig. 10), because of the presence of defect states in BaTaO$_2$N which prevented effective utilization of photoexcited charge carriers.

**Photocatalytic Z-scheme overall water splitting.** Because of the efficient H$_2$-evolution performance of BaTaO$_2$N (RbCl) decorated with Pt by the two-step procedure, photocatalytic Z-scheme water splitting was investigated using this photocatalyst as a HEP. When combined with surface-treated WO$_3$[42,43] as the OEP and IO$_3^-$/I$^-$ as a redox mediator, BaTaO$_2$N loaded with Pt by the two-step decoration method exhibited higher photocatalytic activity in Z-scheme water splitting than Pt-loaded BaTaO$_2$N photocatalysts produced by the individual impregnation–reduction method or the photodeposition method, even though the Pt loading amounts were varied (Fig. 5a). The H$_2$ evolution activities of various Pt-modified BaTaO$_2$N were well correlated in Z-scheme water splitting and half-reactions using sacrificial methanol (Fig. 1a) or NaI aqueous solutions (Supplementary Fig. 14). Furthermore, the H$_2$ evolution rate for various Pt-modified BaTaO$_2$N samples from a NaI solution was lower than the O$_2$ evolution rate of WO$_3$ from a NaIO$_3$ solution (Supplementary Fig. 14), indicating that the rate-determining step for this Z-scheme water splitting is still the generation of H$_2$ on Pt-modified BaTaO$_2$N. Therefore, the improvement in Z-scheme water-splitting activity chiefly resulted from the high quality of the single-crystalline BaTaO$_2$N photo-catalyst, as well as the fine structure of the nanoparticulate Pt cocatalyst afforded by the sequential decoration procedure.

Figure 5b shows the H$_2$ and O$_2$ evolution during Z-scheme water splitting using BaTaO$_2$N decorated with Pt by the two-step procedure as the HEP, surface-treated WO$_3$ as the OEP, and IO$_3^-$/I$^-$ as the redox mediator under visible-light ($\lambda \geq 420$ nm) irradiation. The total amount of gas was calculated from the gas evolution during light irradiation with regular evacuation every 30 min (Supplementary Fig. 15a). Both H$_2$ and O$_2$ were stably evolved at a near-stoichiometric molar ratio of 2:1 without obvious deactivation under visible light. The evolution of N$_2$ was not detected, indicating negligible deterioration of Pt-decorated BaTaO$_2$N over 10 h. The Z-scheme water splitting also occurred stably under simulated sunlight (Fig. 5c and Supplementary Fig. 15b). The STH of this redox-mediated Z-scheme system was 0.24%. It is worth noting that the dependence of Z-scheme water-splitting activity on the concentration of introduced I$^-$ anions under visible light ($\lambda \geq 420$ nm) and under simulated sunlight were not identical (Supplementary Fig. 16). This is because the light excitation conditions influence the carrier dynamics in the photocatalysts and the subsequent redox reaction kinetics on their surfaces[10,44]. The AQY value for this Z-scheme water-splitting system was 4.0% at 420 nm ($\pm 25$ nm) for optimized conditions under monochromatic light (Supplementary Table 6), which is six times higher than that for the previously reported BaTaO$_2$N-based Z-scheme water-splitting system (AQY of 0.6% at $420 - 440$ nm, see Supplementary Table 1)[45]. These AQY and STH values are still behind those of Z-scheme systems constructed with Rh$_y$Cr$_{2-y}$O$_3$-loaded ZrO$_2$-modified TaON (AQY of 10.3% at 420 nm)[10] or Ru-modified SrTiO$_3$:La, Rh (AQY of 33% at 419 nm and STH

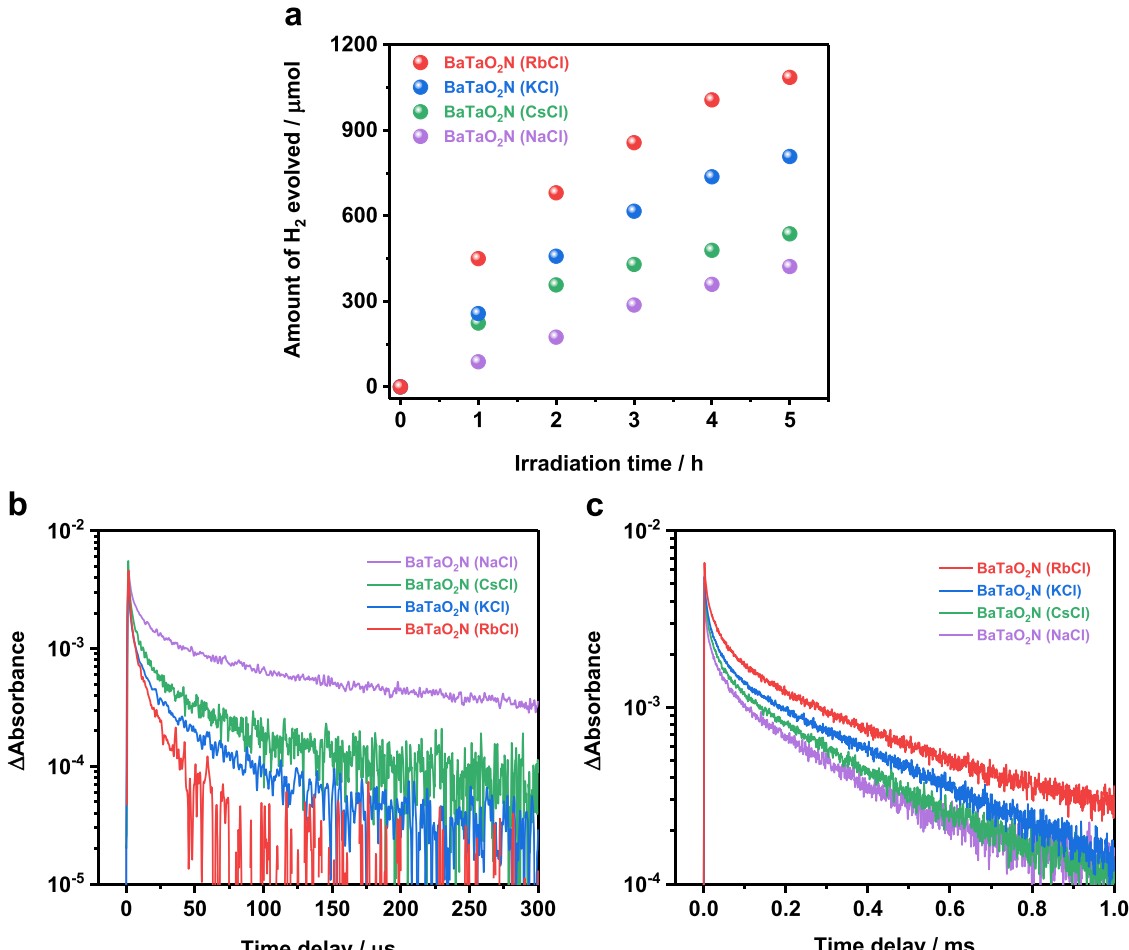

**Fig. 4 Photocatalytic H₂ evolution and photoexcited charge dynamics of Pt-modified BaTaO₂N photocatalysts. a** Photocatalytic performance of $H_2$ evolution on different Pt-modified BaTaO₂N photocatalysts from an aqueous methanol solution. Conditions: Pt-modified BaTaO₂N photocatalyst, 0.1 g; cocatalyst, 0.3 wt% in total (sequential decoration method: 0.1 wt% by impregnation–reduction and 0.2 wt% by photodeposition); 10 vol% aqueous methanol solution, 150 mL; light source, 300 W Xenon lamp equipped with a cut-off filter ($\lambda \geq 420$ nm); reaction system, Pyrex top-illuminated vessel connected to the closed gas-circulation system without evacuation of gas products. **b, c** Transient absorption decays corresponding to electron dynamics (**b**) and hole dynamics (**c**) in different Pt-modified BaTaO₂N photocatalysts probed at 11000 cm$^{-1}$ (910 nm, 1.36 eV) and 15400 cm$^{-1}$ (649 nm, 1.91 eV), respectively.

of 1.1%)[12] as the HEPs (see Supplementary Table 1). We also note that the water-splitting activity of the present Z-scheme system decreased by 30% during continuous illumination for more than 50 h (Supplementary Fig. 17). Moreover, the Z-scheme water-splitting activity dropped with increasing reaction system pressure (Supplementary Fig. 18). This is due in part to rapid water formation from the H₂ and O₂ products (Supplementary Fig. 19), probably on the bare Pt cocatalyst. In addition, the competition between reverse reactions from redox mediators also hinders the intrinsic performance of such Pt-modified BaTaO₂N in the Z-scheme water-splitting system (Supplementary Fig. 20). Thus, improvements in the durability and reaction selectivity, apart from the innovation of photocatalyst preparation and cocatalyst loading protocols, still need to be pursued. Nevertheless, this is by far the most efficient bias-free Z-scheme water-splitting system involving particulate photocatalysts harvesting visible light up to 650 nm. The activation of such 600-nm-class photocatalysts is key to the future development of particulate water-splitting systems. Further improvements in BaTaO₂N-based Z-scheme water-splitting efficiency are expected by refining the preparation of the BaTaO₂N photocatalyst, replacing the WO₃ OEP with a wide wavelength visible-light-harvesting photocatalyst, and exploring effective redox mediators or solid conductive mediators.

In summary, stepwise loading of a Pt cocatalyst by impregnation–reduction and subsequent photodeposition remarkably enhanced the photocatalytic H₂ evolution activity of single-crystalline particulate BaTaO₂N with an absorption edge of 650 nm. The sequential decoration method produced highly dispersed and uniformly sized Pt active sites firmly on BaTaO₂N, enabling the rapid transfer of photogenerated electrons across the interface and an active H₂-evolution reaction on the surface. As a result, the high-quality BaTaO₂N photocatalyst loaded sequentially with a Pt cocatalyst exhibited an AQY of 6.8 ± 0.5% at 420 nm for photocatalytic H₂ evolution from a sacrificial methanol aqueous solution, and an AQY of 4.0% at 420 nm and an STH of 0.24% in Z-scheme water splitting. This is the most efficient solar water-splitting process involving a 600-nm-class particulate photocatalyst without any external bias. Sequential cocatalyst decoration onto single-crystalline oxynitride photocatalysts enables efficient utilization of photoexcited electrons and will contribute to the development of efficient solar-to-chemical energy conversion systems based on narrow-bandgap photocatalysts.

## Methods

**Synthesis of BaTaO₂N particulate photocatalyst.** BaTaO₂N powder was synthesized by flux-assisted one-pot nitridation[24,34]. Ta₂O₅ (99.9%; Kojundo Chemical

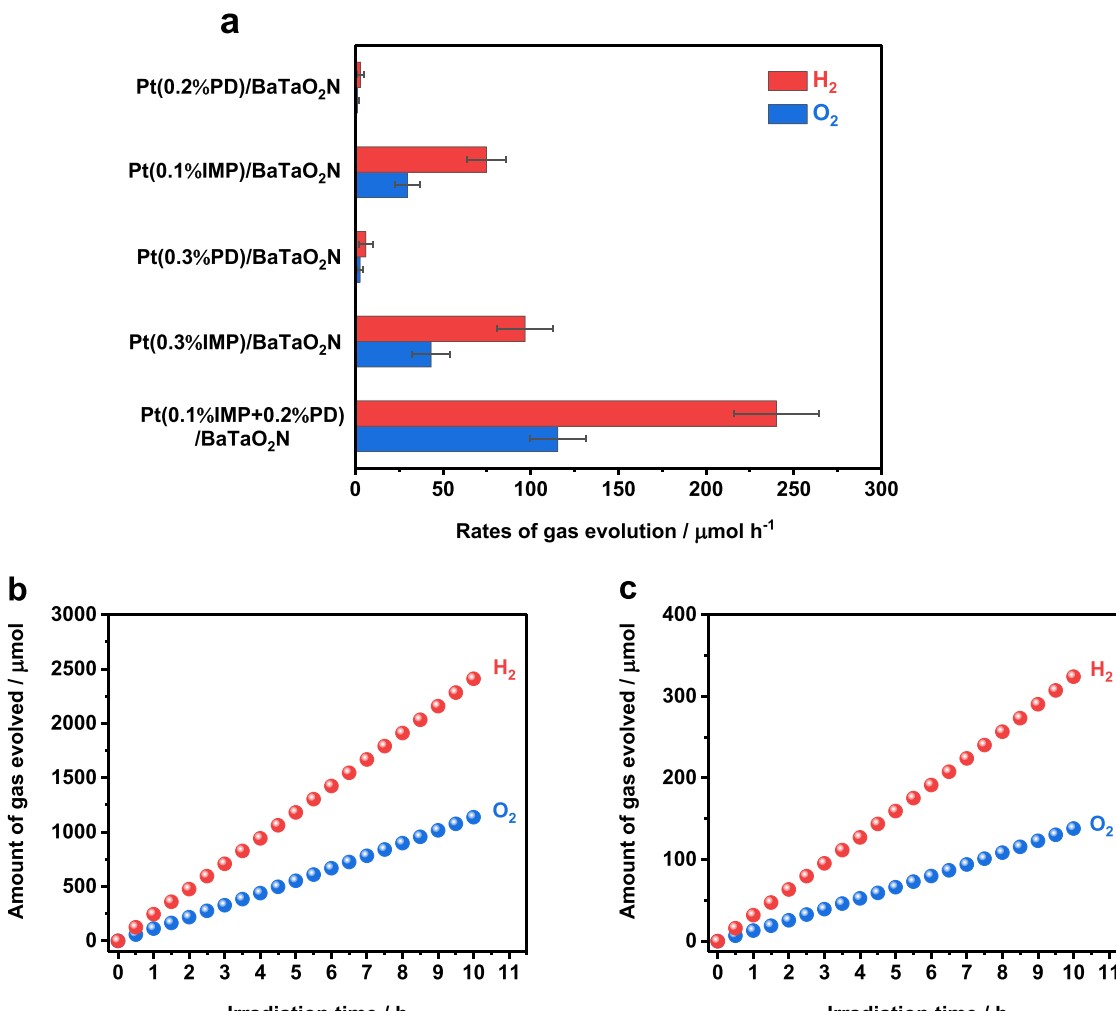

**Fig. 5 Photocatalytic performance of Z-scheme water splitting. a** $H_2$ and $O_2$ evolution rates during Z-scheme water-splitting reaction using Pt-modified BaTaO$_2$N (RbCl) as the HEP. Error bars indicate standard deviation for three measurements. **b**, **c** Time courses of gas evolution during Z-scheme water-splitting reaction using Pt(0.1% IMP + 0.2% PD)/BaTaO$_2$N as the HEP, under visible light ($\lambda \geq 420$ nm) (**b**) and simulated sunlight (**c**). Conditions: Pt-modified BaTaO$_2$N (RbCl) photocatalyst, 0.1 g; surface-treated WO$_3$, 0.15 g; 150 mL aqueous NaI solution, 1 mM for **a** and **b**, and 3 mM for **c**; light source, 300 W Xenon lamp ($\lambda \geq 420$ nm) or solar simulator (AM 1.5 G), irradiation area for solar simulator was 7.6 cm$^2$; reaction system, Pyrex top-illuminated vessel connected to the closed gas-circulation system with the periodical evacuation of gas products.

Laboratory Co., Ltd.) and BaCO$_3$ (99.9%; Kanto Chemical Co., Inc.) were mixed at a Ta:Ba molar ratio of 1:1.1. Excess Ba was added to compensate for losses by volatilization at high temperatures. NaCl (99.0 + %; FUJIFILM Wako Pure Chemical Industries, Ltd.), KCl (99.0 + %; FUJIFILM Wako Pure Chemical Industries, Ltd.), RbCl (95.0 + %; FUJIFILM Wako Pure Chemical Industries, Ltd.) or CsCl (99.0 + %; FUJIFILM Wako Pure Chemical Industries, Ltd.) was added as a flux reagent at a solute concentration of 10 mol%, where the solute concentration is defined as the molar ratio of BaTaO$_2$N to the total amount of BaTaO$_2$N and the flux. After grinding of the reagents in an agate mortar for 30 min, the mixture was transferred into an alumina tube and nitrided at 1223 K for 8 h under a flow of gaseous NH$_3$ at 200 mL min$^{-1}$. The BaTaO$_2$N obtained in this manner was washed with ultrapure water at 343 K for 2 h and filtered thrice to remove any residual flux reagents. The powder was then completely dried at room temperature overnight.

**Modification of BaTaO$_2$N photocatalyst with Pt cocatalyst**. Modification of the BaTaO$_2$N photocatalyst with Pt cocatalysts was conducted by a two-step decoration method. Firstly, a certain amount of Pt cocatalyst was loaded on the BaTaO$_2$N photocatalyst by impregnation followed by H$_2$-reduction treatment. BaTaO$_2$N powder was immersed in an aqueous solution containing the required amount of H$_2$PtCl$_6$·6H$_2$O (>98.5%; Kanto Chemical Co., Inc.) as a Pt precursor. The slurry was continuously stirred with strong sonication for 5 min to completely disperse the BaTaO$_2$N powder in the H$_2$PtCl$_6$ solution. After the slurry was dried in a hot water bath, the resulting powdered mixture was heated at 473 K for 1 h under a flow of mixed H$_2$ and N$_2$ gases (H$_2$: 20 mL min$^{-1}$ and N$_2$: 200 mL min$^{-1}$) to form small metallic Pt grains on the surface of the BaTaO$_2$N photocatalyst. Subsequently, an additional amount of Pt cocatalyst was loaded on the Pt-impregnated

BaTaO$_2$N by the photodeposition process. This was accomplished by dispersing Pt-impregnated BaTaO$_2$N powder in 150 mL of an aqueous methanol solution (10 vol%) containing the required amount of Pt precursor. The pH of this solution was not adjusted and the temperature was maintained at 288 K by circulating cooling water. The suspension was evacuated to completely remove dissolved air and then exposed to visible light ($\lambda \geq 420$ nm) with continuous stirring. The H$_2$ gas evolved during the photodeposition process was detected by gas chromatography as described in the experimental section for photocatalytic H$_2$ production reaction. For comparison, a certain amount of Pt cocatalyst was loaded on pristine BaTaO$_2$N by the individual impregnation–reduction method or the individual photo-deposition method. The procedures for each individual method were the same as in the two-step decoration method. To reveal the location of cocatalyst species deposited during the photodeposition process in the two-step decoration method, Pd particles were loaded on pristine BaTaO$_2$N and Pt-impregnated BaTaO$_2$N photocatalysts by photodeposition for 1 h using PdCl$_2$ (99.9%; FUJIFILM Wako Pure Chemical Industries, Ltd.) as the precursor. The introduced Pd amount was 0.2 wt% with respect to the BaTaO$_2$N photocatalysts. Moreover, pristine BaTaO$_2$N powder was subjected to reduction treatment at 473 K for 1 h under a flow of mixed H$_2$ and N$_2$ gases (H$_2$: 20 mL min$^{-1}$ and N$_2$: 200 mL min$^{-1}$) without the introduction of Pt species. Then Pt particles were loaded on the H$_2$-treated BaTaO$_2$N by the same photodeposition method.

**Surface treatment of WO$_3$ photocatalyst**. Surface-treated WO$_3$ for Z-scheme water splitting was prepared in the following procedure[42,43]. PtO$_x$ (0.5 wt% as Pt) was loaded on WO$_3$ (99.99%, Kojundo Chemical Laboratory Co., Ltd.) by immersing WO$_3$ powder in an aqueous H$_2$PtCl$_6$ solution, followed by calcination

in air at 823 K for 0.5 h. The obtained PtO$_x$-loaded WO$_3$ was then impregnated in a Cs$_2$CO$_3$ (97%, FUJIFILM Wako Pure Chemical Industries, Ltd.) solution with a molar ratio of Cs to W of 1%. After calcination in air at 773 K for 10 min, the Cs$^+$-treated and PtO$_x$-loaded WO$_3$ were soaked in 1 M H$_2$SO$_4$ solution with vigorous stirring for 1 h in order to adequately induce ion-exchange reactions. Finally, the resulting powder was collected by filtration and dried in air at room temperature overnight.

**Characterization of material**. X-ray diffraction (XRD) patterns were acquired using a Rigaku MiniFlex 300 powder diffractometer with Cu Kα radiation, operating at 30 kV and 30 mA. UV–vis diffuse reflectance spectra (DRS) were recorded with a spectrophotometer (V-670, JASCO) equipped with an integrating sphere, with a Spectralon standard as a reference for baseline correction. Scanning electron microscopy (SEM) images were obtained on the Hitachi SU8020 system and JEOL JSM-7500FA. High-resolution transmission electron microscopy (HRTEM) and energy-dispersive X-ray spectroscopy (EDS) were conducted with a JEM-2800 system (JEOL) and an X-MAX 100TLE SDD detector (Oxford Instruments). The binding energies were determined by X-ray photoelectron spectroscopy (XPS) on a PHI Quantera II spectrometer with an Al Kα X-ray source and normalized to C 1 *s* for each sample. The elemental analysis was performed by inductively coupled plasma-atomic emission spectroscopy (ICP-AES, Thermo Fischer Scientific, iCAP 7600duo) and oxygen/nitrogen/hydrogen (ONH) analysis (LECO Corporation, TCH600).

Microsecond transient absorption (TA) measurements were performed using a Nd:YAG laser system (Continuum, Surelite I; duration: 6 ns) with custom-built spectrometers[35]. The IR probe light emitted from the MoSi$_2$ coil was focused on the sample and then the transmitted light was introduced to a grating spectrometer, which allowed measurement of probe energies from 6000 cm$^{-1}$ (1667 nm, 0.74 eV) to 1000 cm$^{-1}$ (10 μm, 0.12 eV). The monochromated light was detected by a mercury cadmium telluride (MCT) detector (Kolmar). For the visible and NIR region from 20000 cm$^{-1}$ (500 nm, 2.47 eV) to 6000 cm$^{-1}$ (1667 nm, 0.74 eV), the measurements were carried out in reflection mode, i.e., the reflected light from the sample entered the grating spectrometer and was then detected by Si photodetectors. The output electric signal was amplified with an AC-coupled amplifier (Stanford Research Systems, SR560, 1 MHz), which can measure responses on a timescale of one microsecond to milliseconds. Laser pulses (480 nm, 3 mJ pulse$^{-1}$) were used to excite pristine BaTaO$_2$N and Pt-loaded BaTaO$_2$N photocatalysts via bandgap transitions. The time resolution of the spectrometer was limited to 1 μs by the response of photodetectors. The output electric signal was amplified using AC-coupled amplifier with a bandwidth of 1 MHz, which can measure responses in the timescale of one microsecond to milliseconds. One thousand responses were accumulated to obtain the intensity trace at a single wavenumber or a decay curve. In order to rule out thermal effects or IR emission, the absorption spectra and absorbance changes were determined after subtracting the measurements without probe light. The experiments were performed in a vacuum at room temperature.

**Photocatalytic reactions of H$_2$ evolution, O$_2$ evolution, and Z-scheme water splitting**. Photocatalytic H$_2$ evolution reactions were carried out in a Pyrex top-illuminated reaction vessel connected to a closed gas-circulation system. A Pt-loaded BaTaO$_2$N photocatalyst was dispersed in 150 mL of aqueous methanol solution or aqueous NaI solution. The pH of this solution was not adjusted and the temperature was maintained at 288 K by circulating cooling water. After completely removing air from the reaction slurry by evacuation, the suspension was irradiated with a 300 W Xenon lamp equipped with a cold mirror and a cut-off filter (L42, λ ≥ 420 nm). The reactant solution was maintained at 288 K by a cooling water system during the reaction. The evolved gas products were analyzed using an integrated thermal conductivity detector-gas chromatography system (TCD–GC) consisting of a GC-8A chromatograph (Shimadzu Corp.) equipped with a Molecular Sieve 5 Å column, with argon as the carrier gas. The sensitivity of the TCD was calibrated by analyzing known amounts of gas introduced into the fully evacuated reaction system containing reaction solutions under illumination. For the O$_2$ evolution reaction, surface-treated WO$_3$ was dispersed in 150 mL of aqueous NaIO$_3$ solution (20 mM). The reaction trial was performed in the same system with the same procedure as for the hydrogen-evolution reactions.

Z-scheme water-splitting reactions were carried out in a Pyrex top-illuminated reaction vessel connected to a closed gas-circulation system or a gas-flow system. Pt-loaded BaTaO$_2$N as the HEP and surface-treated WO$_3$ as the OEP was dispersed in 150 mL of an aqueous solution containing NaI at a certain concentration. The pH of the solution was not adjusted. After completely removing air from the reaction slurry by evacuation, the suspension was irradiated by a 300 W Xenon lamp equipped with a cold mirror and a cut-off filter (L42, λ ≥ 420 nm) or by a solar simulator (SAN-EI electronic, XES40S1, AM 1.5 G, 100 mW cm$^{-2}$). For the reaction under simulated sunlight, the top window of the reaction vessel was covered with a mask to confine the irradiated sample area to 7.6 cm$^2$. The reactant solution was maintained at 288 K by a cooling water system during the reaction. The reaction system was periodically evacuated at an interval of 30 min. The gas products evolved during each 30 min irradiation period were analyzed using the integrated TCD–GC. In the experiments using a gas flow system, a designated amount of Ar gas was fed to the reaction suspension, and the reaction suspension

was evacuated with a dry pump (ULVAC, DOP-40D) through a vacuum regulator (Koganei, NVR200) and a sampling valve (GL Sciences, AU-CF-6J) that was controlled by a remote timer (GL Sciences, RT731A). The pressure of the suspension was monitored using a vacuum gauge inserted just upstream of the reactor. The gas products were detected by the integrated TCD–GC that was calibrated at the respective pressures.

**Solar-to-hydrogen conversion efficiency measurements**. The water-splitting reaction was performed under simulated solar irradiation. The solar-to-hydrogen (STH) conversion efficiency is given by Eq. (1):

$$STH(\%) = (R(H_2) \times \Delta G_r)/(P \times S) \times 100 \qquad (1)$$

where $R(H_2)$, $\Delta G_r$, $P$, and $S$ denote the rate of hydrogen evolution during the Z-scheme overall water-splitting reaction, the Gibbs energy for the reaction H$_2$O(l) → H$_2$(g) + 1/2O$_2$(g), the energy intensity of the AM1.5 G solar irradiation (100 mW cm$^{-2}$), and the irradiated sample area (7.6 cm$^2$), respectively. The value of $\Delta G_r$ used for the calculations was 237 kJ mol$^{-1}$ at 288 K. Because the O$_2$ evolution rate was slightly deficient compared to that for stoichiometric water splitting, $R(H_2)$ used in this STH calculation was the average of the H$_2$ evolution rate and twice the O$_2$ evolution rate.

**Apparent quantum yield measurement**. The apparent quantum yield (AQY) for photocatalytic reaction is given by Eq. (2):

$$AQY(\%) = [A \times R]/I \times 100 \qquad (2)$$

where $R$ and $I$ represent the rate of gas evolution and the incident photon flux, respectively. $A$ is the number of electrons consumed to generate one molecule of H$_2$, and is 2 for photocatalytic hydrogen production from sacrificial methanol solution, and 4 for Z-scheme water splitting based on two-step photoexcitation. The photocatalytic reaction and measurement of the number of incident photons were carried out using the same light source equipped with various band-pass filters. The number of incident photons illuminating the reaction cell was measured using a grating spectroradiometer.

## Data availability

The data that support the findings of this study are available from the corresponding author upon reasonable request. Source data are provided with this paper.

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

## Acknowledgements

This work was financially supported by the Artificial Photosynthesis Project of the New Energy and Industrial Technology Development Organization (NEDO). A part of this work was conducted at the Advanced Characterization Nanotechnology Platform of the University of Tokyo, supported by the Nanotechnology Platform of the Ministry of Education, Culture, Sports, Science, and Technology (MEXT), Japan (grant number: JPMXP09A-19-UT-0023). The authors thank Ms. Michiko Obata of Shinshu University for her assistance with XPS measurements.

## Author contributions

Z.W., Y.L., K.T. and K.D. designed the research. Z.W. and Y.L. conducted the photocatalyst material fabrications, cocatalyst modifications, photocatalytic $H_2$ production reactions, Z-scheme overall water-splitting reactions, XRD, UV–vis DRS, and SEM characterizations. Z.W. and T.H. conducted XPS measurements. J.M.V. and A.Y. carried out the TAS experiments and analyzed the resulting data. Z.W., Y.L., S.C. and L.L. performed AQY and STH measurements. M.N and N.S. conducted HRTEM and HRTEM-EDS measurements. N.K. carried out ICP-AES measurements. T.H., S.S., K.T. and K.D. supervised the entire research work. Z.W., Y.L, T.H., J.M.V., S.S., S.C., Z.P., T.T., K.T. and K.D. discussed the results. Z.W., Y.L, T.H., K.T. and K.D. wrote and revised the paper with contributions from the other authors.

## Competing interests

The authors declare no competing interests.
