## [Peer Review File · Nature Communications]

REVIEWER COMMENTS

Reviewer #1 (Remarks to the Author):

This report on the effect of flux treatment and Pt deposition on solar hydrogen generation with a barium tantalum oxynitride is novel, interesting, and has strong scientific merit. The work was done in an expertly manner and the findings are well described in a concise and clear fashion, with high quality figures. I recommend publication in Nat. Comm. after minor revision.

1. There are significant reactivity differences depending on the nature of the flux reagent. Can the authors speculate on whether these differences arise from the different shapes of the particles, or if incorporation of the flux reagent into the BaTaO₂N lattice plays a role.
2. The authors use several kinds of photoreactors to test their catalysts, but it is not always evident from the figure caption, which reactor and mode of gas evacuation was used in each case. For example, caption Fig 5 does not contain this information. Please add the info to each caption.
3. The optical appearance of a photocatalyst can provide important information about the presence of oxygen vacancies in the material. The authors should provide photos of their champion materials.

Sincerely yours,
Frank Osterloh

Reviewer #2 (Remarks to the Author):

My recommendation is to publish this paper, after minor corrections. The key result of this paper is that using a novel 2 step deposition for the cocatalyst it is possible to significantly increase the hydrogen generation rate. It has been widely shown that for the small bandgap visible light absorbing photocatalysts the use of co-catalysts are essential for enhancing activity. Therefore, this report showing how to best optimise the co-catalyst will be of great interest to the photocatalytic community. The introduction briefly sets out the prior work in the area and highlights the key results. When discussing the preparation of the BaTaO₂N, the paper states that the full experimental details can be found in the ESI. However they seem to be at the end of the paper, rather than in the ESI. This should be clarified. For the main result convincing arguments are made for why the 2 step deposition leads to enhanced hydrogen production and a plausible mechanism, and this is fully supported by the materials characterisation TEM, SEM and TAS measurements.

Sufficient materials characterisation of the BaTaO₂N (RbCl) is provided, but the paper also provides a secondary result comparing the effect of the different alkali chloride fluxes in the synthesis of BaTaO₂N. This shows that RbCl leads to the highest activity – however the reason for this is not clearly presented in the paper – principally due to insufficient materials characterisation of the other samples. The result is important and worthy of dissemination, but it is necessary to provide more evidence explaining the difference in the samples.

In the final section comparing the overall water splitting rates between previously reported systems (p12) I would like to see the comparison being made quantitatively rather than qualitatively. On a point of style, it would be easier to review the manuscript if the figures were presented in main body of the paper near where they are discussed. The placing of the figures at the end of the paper is an outdated convention that inconveniences the reviewer with little other benefit.

Reviewer #3 (Remarks to the Author):

The authors have prepared an interesting photocatalytic material (BaTaO₂N) able to form molecular hydrogen from the dehydrogenation of methanol upon illumination at 420 nm (that is slightly above

the UV(A) region) with a considerable quantum efficiency and to even split water in a Z-scheme system when in contact with surface modified WO₃. This work will be of interest to those working in the field and its acceptance for publication is therefore suggested following some revision as indicated below.

- 1) Is any molecular hydrogen being formed in the absence of any sacrificial agent?
- 2) All photocatalytic experiments are performed in evacuated systems, please present some data obtained under ambient pressure!
- 3) When characterized after having been used as photocatalysts are the powders still the same? Does this also include the TAS data?

Response to Reviewers' Comments

Manuscript number: NCOMMS-20-40366-T

Manuscript type: Research Article

Title: Sequential cocatalyst decoration on BaTaO₂N towards highly-active Z-scheme water splitting

Correspondence Authors: Prof. Kazunari Domen, Prof. Katsuya Teshima

Authors: Zheng Wang, Ying Luo, Takashi Hisatomi, Junie Jhon M. Vequizo, Sayaka Suzuki, Shanshan Chen, Mamiko Nakabayashi, Lihua Lin, Zhenhua Pan, Nobuko Kariya, Akira Yamakata, Naoya Shibata, Tsuyoshi Takata, Katsuya Teshima and Kazunari Domen

[The reviewer comments are shown in *italic*; responses are in **blue**; all revisions in the main text and SI are highlighted in **red**]

Reviewer 1:

General comment: *This report on the effect of flux treatment and Pt deposition on solar hydrogen generation with a barium tantalum oxynitride is novel, interesting, and has strong scientific merit. The work was done in an expertly manner and the findings are well described in a concise and clear fashion, with high quality figures. I recommend publication in Nat. Comm. after minor revision.*

Response: We are very grateful to the Reviewer for the very positive comments and the recommendation for publication of our work in Nature Communications. We also appreciate the Reviewer's valuable suggestions that have helped us to improve our paper. We have taken all comments and suggestions into account. The response to each question is described below.

Comment 1: *There are significant reactivity differences depending on the nature of the flux reagent. Can the authors speculate on whether these differences arise from the different shapes of the particles, or if incorporation of the flux reagent into the BaTaO₂N lattice plays a role?*

Response 1: We agree with the Reviewer that the photocatalytic activities of different BaTaO₂N materials modified by sequential cocatalyst decoration method depend on the nature of BaTaO₂N materials prepared with different flux reagents. The H₂ evolution activities of the different BaTaO₂N photocatalysts decrease in this order: BaTaO₂N (RbCl), BaTaO₂N (KCl), BaTaO₂N (CsCl) and BaTaO₂N (NaCl) (Fig. 4a). However, the four flux-assisted BaTaO₂N photocatalysts exhibited similar cuboid shape except for BTON (CsCl) having some small steps at the edges of cuboid structure (Fig. 2a and Supplementary Fig. 10). In addition, we added ICP-AES

analysis for different BaTaO₂N materials (Supplementary Table 5). It is found that some alkali metal ions were incorporated in the materials through the flux-assisted nitridation process, and that the incorporation amount decreased with the increase of their ionic radii. We think that the incorporation of alkali metal ions would induce charge imbalance and defects in the BaTaO₂N material, thus leading to less efficient photoexcited charge carrier transfer.

Moreover, through the comparison of XRD pattern (Supplementary Fig. 13a), background absorption of UV-vis DRS (Supplementary Fig. 13b) and XPS analysis (Supplementary Fig. 11) for these four BaTaO₂N materials, it is also found that the crystallinity and defect densities (reduced Ta⁵⁺ species) of BaTaO₂N vary with the application of different flux reagent in the synthesis. BaTaO₂N synthesized with RbCl flux has better crystallinity and lower defect densities, which also accounts for the best photocatalytic activity.

The morphology, ion incorporation, the crystallization and the formation of defect states for BaTaO₂N materials prepared by flux-assisted nitridation method play important roles in the photocatalytic performance, as has been also discussed in our previous works (*ACS Appl. Mater. Interfaces* **11**, 22264–22271 (2019), *Cryst. Growth Des.* **20**, 255–261 (2020)). It is generally difficult to predict the activity from single feature of the photocatalyst. However, based on the various analytical results, it is reasonable to conclude that the RbCl flux-assisted nitridation method results in the high quality of BaTaO₂N materials, which take full advantage of two-step Pt cocatalyst decoration for the enhancement of photoexcited charge transfer towards efficient H₂ production. The related information has been included in Page 14 and 15 of revised manuscript and Supplementary Fig. 11 and 13 and Supplementary Table 5.

Comment 2: *The authors use several kinds of photoreactors to test their catalysts, but it is not always evident from the figure caption, which reactor and mode of gas evacuation was used in each case. For example, caption Fig 5 does not contain this information. Please add the info to each caption.*

Response 2: We thank the Reviewer for pointing out this issue. We have added the information of reaction system and mode of gas evacuation in the captions of Fig. 1, 4 and 5, Supplementary Fig. 2, 14–18 and 20, and Supplementary Table 6. For your information, all of the photocatalytic reactions were carried out using the essentially identical Pyrex glass reactor with the Pyrex glass window for top illumination. The reactor was set up in a closed gas-circulation system or a gas-flow system with the connection to an evacuation pump. Before the photocatalytic reaction, the reaction solution and system were evacuated completely to remove air. During the photocatalytic reaction, the gas products were accumulated in the closed gas-circulation system, whereas the gas products were continuously swept by Ar flow in the gas-flow system.

Comment 3: The optical appearance of a photocatalyst can provide important information about the presence of oxygen vacancies in the material. The authors should provide photos of their champion materials.

Response 3: We thank the reviewer for the valuable suggestion. We have included the optical photos for BaTaO₂N materials synthesized with the different flux reagents, together with their UV-vis DRS in Supplementary Fig. 13b, and the brief discussion about oxygen vacancy in Page 14 of revised manuscript. All the samples show the light absorption edge at 650 nm with similar brownish-red colours, which is typical optical feature for BaTaO₂N.

Reviewer 2:

General comment: My recommendation is to publish this paper, after minor corrections. The key result of this paper is that using a novel 2 step deposition for the cocatalyst it is possible to significantly increase the hydrogen generation rate. It has been widely shown that for the small bandgap visible light absorbing photocatalysts the use of co-catalysts are essential for enhancing activity. Therefore, this report showing how to best optimise the co-catalyst will be of great interest to the photocatalytic community.

Response: We are very appreciative of the Reviewer's recommendation for publication and positive evaluation of our work. We also thank the Reviewer for the constructive comments that help us to enrich our paper. We have considered all of the points that were raised. The response to each question is described below.

Comment 1: The introduction briefly sets out the prior work in the area and highlights the key results. When discussing the preparation of the BaTaO₂N, the paper states that the full experimental details can be found in the ESI. However they seem to be at the end of the paper, rather than in the ESI. This should be clarified.

Response 1: We thank the reviewer for pointing out the inappropriate statement. We have changed that "The preparation of BaTaO₂N, loading of Pt nanoparticles on BaTaO₂N, and the photocatalytic H₂ production and Z-scheme water splitting, are described in detail in the section of Methods". The revised sentence is shown in Page 5.

Comment 2: Sufficient materials characterization of the BaTaO₂N (RbCl) is provided, but the paper also provides a secondary result comparing the effect of the different alkali chloride fluxes in the synthesis of BaTaO₂N. This shows that RbCl leads to the highest activity – however the reason for this is not clearly presented in the paper – principally due to insufficient materials characterization of the other samples. The result is important and worthy of dissemination, but it is necessary to provide more

evidence explaining the difference in the samples.

Response 2: We greatly appreciate the Reviewer's valuable suggestions. In order to explain the different activities on the BaTaO₂N photocatalysts synthesized with different alkali chloride fluxes, we compared XRD patterns, UV-vis DRS, XPS and ICP-AES analysis and found clear differences among four photocatalysts, as reported in our previous works (*ACS Appl. Mater. Interfaces* **11**, 22264–22271 (2019), *Cryst. Growth Des.* **20**, 255–261 (2020)). XRD patterns (Supplementary Fig. 13a) show that the crystallinity of RbCl-assisted BaTaO₂N was higher than that of the other flux samples, and the background absorption of UV-vis DRS (Supplementary Fig. 13b) and XPS analysis (Supplementary Fig. 11) indicate the minimized defect densities (reduced Ta⁵⁺ species and anion vacancies) in BaTaO₂N synthesized with RbCl flux. Moreover, it is found that some alkali metal ions were incorporated in the materials through the flux-assisted nitridation process, and that the incorporation amount decreased with the increase of their ionic radii (Supplementary Table 5). The incorporation of alkali metal ions would induce charge imbalance and defects in the BaTaO₂N material, thus leading to less efficient photoexcited charge carrier transfer. In addition, XPS analysis also reflects that the existence of flux ion in the top surface layers was negligible for BaTaO₂N synthesized with RbCl, KCl or CsCl flux and Pt cocatalysts were loaded as the metallic state with the same content (Supplementary Fig. 11). Considering the above characterizations together with TAS results (Fig. 4b and 4c and Supplementary Fig. 12), the densities of structural defects and mid-gap states affect the dispersion of Pt cocatalysts on these flux-assisted BaTaO₂N samples despite the similarity in crystal shapes (Fig. 2a and Supplementary Fig. 10), which in turn affects the utilization of photoexcited charges. Therefore, we conclude that the photocatalytic performance for H₂ evolution is dependent on the quality of BaTaO₂N materials, and the high quality of RbCl-assisted BaTaO₂N photocatalyst can take full advantage of the sequential cocatalyst decoration method. The related information is included in Page 14 and 15 of revised manuscript and Supplementary Fig. 11 and 13 and Supplementary Table 5.

Comment 3: *In the final section comparing the overall water splitting rates between previously reported systems (p12) I would like to see the comparison being made quantitatively rather than qualitatively.*

Response 3: We have listed the gas evolution rates, apparent quantum efficiency (AQY) and solar-to-hydrogen conversion efficiency (STH) of the previously reported Z-scheme water splitting systems in Supplementary Table 1. It is difficult to compare gas evolution rates for different systems, because the irradiation conditions such as light intensity and irradiation area are different. Therefore, we added the champion values of AQY and STH for quantitative comparison in Page 18 of revised manuscript (Page 12 of previous version)

Comment 4: *On a point of style, it would be easier to review the manuscript if the*

figures were presented in main body of the paper near where they are discussed. The placing of the figures at the end of the paper is an outdated convention that inconveniences the reviewer with little other benefit.

Response 4: We are sorry for making the Reviewer inconvenience in reviewing our manuscript. We have relocated the figures of main text near the related discussion part in the revised manuscript.

Reviewer 3:

General comment: *The authors have prepared an interesting photocatalytic material (BaTaO₂N) able to form molecular hydrogen from the dehydrogenation of methanol upon illumination at 420 nm (that is slightly above the UV(A) region) with a considerable quantum efficiency and to even split water in a Z-scheme system when in contact with surface modified WO₃. This work will be of interest to those working in the field and its acceptance for publication is therefore suggested following some revision as indicated below.*

Response: We are very thankful for the Reviewer's positive evaluation of our work and valuable advices that improve the quality of our manuscript. We have carefully considered all of the questions that were raised. The response to each question is described below.

Comment 1: *Is any molecular hydrogen being formed in the absence of any sacrificial agent?*

Response 1: There will be no hydrogen gas evolved on BaTaO₂N photocatalyst in the absence of any sacrificial agent. If hydrogen gas can be generated on BaTaO₂N photocatalyst without any sacrificial agent, the oxygen evolution should take place by utilizing water molecule as the electron donor. However, we did not observe such phenomenon at present. Coloadng of hydrogen evolution cocatalysts and oxygen evolution cocatalysts is essential to evolve hydrogen and oxygen simultaneously using BaTaO₂N, but this necessitates a lot of optimization studies and is beyond the scope of this work. Instead, we studied the Z-schematic water splitting using the developed Pt-loaded BaTaO₂N as the hydrogen evolution photocatalyst to demonstrate more efficient water splitting than many other studies.

Comment 2: *All photocatalytic experiments are performed in evacuated systems, please present some data obtained under ambient pressure!*

Response 2: We thank the Reviewer for raising the important issue. We have shown the effect of background pressure on the Z-scheme water splitting activity in Supplementary Fig. 18 and the discussion in Page 18. Because the closed

gas-circulation system can be only operated at reduced pressure, we used gas-flow system to check the photocatalytic performance under ambient pressure. The evolution rates for H₂ and O₂ at ambient pressure were 31.7 and 17.0 μmol h⁻¹, respectively. Please see Supplementary Fig. 18b. It is shown that the Z-scheme water splitting activity dropped with increasing reaction system pressure, probably due to rapid water formation from the H₂ and O₂ products (Supplementary Fig. 19) on the bare Pt cocatalyst. Nevertheless, further enhancement of the photocatalytic activity of this Z-scheme system and elimination of system pressure effect are now ongoing in our investigation.

***Comment 3:** When characterized after having been used as photocatalysts are the powders still the same? Does this also include the TAS data?*

Response 3: We thank the Reviewer for pointing out this important question. Because it is very difficult to recycle the Pt-modified RbCl-assisted BaTaO₂N powder from the mixture of BaTaO₂N and WO₃ photocatalysts after the Z-scheme water splitting reaction, we carried out XRD, XPS and SEM measurements (Supplementary Fig. 3) for the Pt-modified RbCl-assisted BaTaO₂N photocatalyst recycled after a half reaction in the methanol solution. Through comparison of XRD patterns, XPS analysis and SEM images, it is found that the crystallinity and surface compositions of BaTaO₂N material and the loading content, chemical state and dispersion of Pt cocatalysts remained unchanged after a photocatalytic reaction in the methanol solution for 5 h, indicating Pt-modified RbCl-assisted BaTaO₂N photocatalyst was stable in the photocatalytic reaction in this time scale. The related information was included in Page 8 of revised manuscript and Supplementary Fig. 3. On the other hand, it is not possible to perform TAS measurement for the recycled sample in a reasonable time due to the pandemic of COVID-19.

REVIEWERS' COMMENTS

Reviewer #1 (Remarks to the Author):

I am happy with the revisions and recommend publication as is.

Reviewer #2 (Remarks to the Author):

After the minor amendments I am now satisfied that the paper meets the requirements of the journal, and would like to recommend publication.

Reviewer #3 (Remarks to the Author):

The authors have attended all comments from the 3 reviewers adequately. Therefore, acceptance of this manuscript for publication in Nature Communications can be suggested.